# Hybrid SARS-CoV-2 immunity among frontline workers in a high-tourism setting: A community-based serosurvey in Ambergris Caye, Belize, June 2022

Oluwadara Okeremi[1,2,3☯], Ella Hawes[1,2,3☯], Anh N. Ly[1,2,3], Russell Manzanero[4], Sarah M. Gunter[1,2,3], Shannon E. Ronca[1,2,3], Allison Lino[1,2,3], Allyson Hidalgo[1,2,3], Meghan Si[1,2,3], Allison Stewart-Ruano[5], Adriana Maliga[1,2,3], Eric J. Nilles[6], Gerhaldine Morazan[1,2,3], Kristy O. Murray[7☯*]

1 Department of Pediatrics, Baylor College of Medicine and Texas Children's Hospital, Houston, Texas, United States of America, 2 National School of Tropical Medicine, Baylor College of Medicine, Houston, Texas, United States of America, 3 William T. Shearer Center for Human Immunobiology, Texas Children's Hospital, Houston, Texas, United States of America, 4 Belize Ministry of Health and Wellness, Belmopan, Belize, 5 Columbia University, New York City, New York, United States of America, 6 Brigham and Women's Hospital, Harvard University, Boston, Massachusetts, United States of America, 7 Department of Pediatrics, Emory University School of Medicine and Children's Healthcare of Atlanta, Atlanta, Georgia, United States of America

☯ These authors contributed equally.
* kristy.murray@emory.edu

## Abstract

Tourism is central to Belize's economy, yet the influx of international travelers may contribute to infectious disease introduction and transmission among local workers, including SARS-CoV-2. In June 2022, we conducted a cross-sectional seroepidemiologic study among tourism and government workers in the tropical island tourism center of Ambergris Caye, Belize. The goal of this study was to determine SARS-CoV-2 antibody prevalence for both vaccination (anti-spike protein) and natural infection (anti-nucleocapsid protein), vaccine uptake, history of COVID-like symptoms, and risks for infection two years into the pandemic. Workers from 30 hotels and government agencies were enrolled. Participants completed questionnaires to capture demographics, medical history, COVID-19 exposures, and vaccination history. Serum samples were tested for SARS-CoV-2 anti-spike IgG (indicative of vaccination and/or infection) and anti-nucleocapsid IgG (infection only). Of 551 participants, 428 (78%) were tourism workers and 123 (22%) government employees. COVID-19 vaccine uptake was high (98%), and anti-spike IgG seroprevalence was 99%; only three vaccinated participants were seronegative. Anti-nucleocapsid prevalence was 81%, indicating a high prevalence of past natural infection. Based on multivariable analysis, nucleocapsid positivity was independently associated with having ≥5 people in household (aOR=2.08, p = 0.018), while receiving a COVID-19 booster was protective (aOR=0.57, p = 0.013). Among 282 participants reporting previous COVID-19 or

**Data availability statement:** Data have been deposited into a public data repository (Harvard Dataverse): https://doi.org/10.7910/DVN/BX5HUU.

**Funding:** This study was generously supported through a cooperative agreement with the United States Centers for Disease Control and Prevention (U01 GH002235 to KOM). The funders were consulted on the design of the study but had no role in data collection and analysis, decision to publish, or preparation of the manuscript.

**Competing interests:** The authors have declared that no competing interests exist.

COVID-19-like illness, 46% sought medical care and eight were hospitalized. High seroprevalence of both vaccine- and infection-induced antibodies was observed, highlighting the emergence of hybrid immunity among frontline workers. These findings underscore the importance of continued surveillance and targeted public health interventions in tourism-dependent settings.

## Introduction

Low-and-middle-income countries (LMICs) have a high risk for transmission of emerging pathogens due to large population centers, areas of poverty, tropical climates and high population movement. Belize is a small Central American country located on the Caribbean coast, and while considered upper-middle income, more than one-third of the population lives in poverty [1]. Tourism is the largest source of foreign exchange in Belize's economy [1, 2] presenting an additional factor that may contribute to disease introduction. In 2019, > 500,000 tourists visited Belize [3].

These factors put Belize at significant risk for SARS-CoV-2 transmission and led to strict control measures shortly after the first COVID-19 case was detected in March 2020. Measures included prompt closing of borders to international visitors despite the negative economic consequences [4, 5]. The reopening of borders in October 2020 and relaxation of entry criteria [6, 7] posed exposure risks among those working in the tourism and government sectors. To protect the health of workers and travelers, the Belize Ministry of Health and Wellness (MOHW) and Belize Tourism Board created mandatory guidelines for improved sanitation and disease prevention in hotels and tourism areas [6]. Additionally, front line workers, including tourism and government workers, were prioritized during Belize's COVID-19 vaccination campaigns. Despite these targeted prevention measures, the high volume of international visitors and surges in cases (https://covid19.who.int/data) both before and after the vaccination campaigns continued to raise concerns over infection risk for workers whose jobs required continuous interaction with international visitors. Hybrid immunity derived from both vaccination and natural infection can help reduce the severity of COVID-19 infections, even with exposures to novel, emerging variants [8, 9]; however, the uptake of vaccination and true burden of natural infections were not well known in Belize, including high tourism regions where the risk was even greater.

To assess this risk of infection and to develop an understanding of hybrid immunity, we evaluated the seroprevalence of SARS-CoV-2 among front-line, public-facing tourism and government workers in Ambergris Caye, Belize, a high-density international tourist destination. The objectives of this study were to (1) measure the seroprevalence of antibodies to SARS-CoV-2 spike protein (vaccination and/or natural infection) and nucleocapsid protein (natural infection only) by tourism sector and government workers, (2) assess history of exposure to COVID-19 and symptoms of COVID-19-like illness, and (3) determine risk factors for infection. Characterizing the risk among public-facing personnel is critical for informing the response to future epidemic and pandemic threats.

## Materials and methods

This cross-sectional sero-epidemiological study approach was based on the World Health Organization (WHO) UNITY protocol for COVID-related investigations, which was developed to provide a standardized method for assessing and comparing seropositivity between countries [10]. We adapted this protocol for Belize to systematically collect epidemiological exposure data and biological samples.

### Ethics statement

This protocol was approved by the BCM Institutional Review Board (H-50470) and the Belize MOHW. The study was conducted in collaboration with the Belize Ministry of Health, and the Mayor of San Pedro town in Ambergris Caye reviewed the protocol, provided verbal permission for the study to take place, and assisted with sensitizing the community to the goals of the study. The consent form was read aloud to prospective participants prior to inclusion based on their language of preference (English or Spanish). Those who preferred Spanish were consented and interviewed by native Spanish speakers. All study participants provided written informed consent which was documented and witnessed in-person at the time of enrollment by study personnel. Minors under the age of 18 years were not included in the study.

### Study Population and Recruitment

Our target population in Ambergris Caye was: (1) hotel/restaurant employees in the tourism sector, and (2) government workers, including first responders (police and fire), teachers, and Town Council staff. Our sample size assumptions for seropositivity among hotel workers were estimated to be 22%, with a precision of 4% (meaning the true seroprevalence would be between 18% and 26%), a confidence interval of 95% with 80% power, and a design effect of 1.5. This gave us a targeted sample size of 500. The government sector had only 180 employees; therefore, all were eligible to participate.

We identified hotels in Ambergris Caye with >10 rooms (n = 48) and estimated that ~1,700 hospitality employees worked at eligible hotels. To select hotels for sampling, we generated a numbered list of all eligible hotels and assigned clusters to each hotel based on the number of employees, resulting in a unique ID for each hotel/cluster combination. We used a random number generator to determine which cluster and, in turn, which hotel to enroll. Using this approach, 27 hotels were randomly selected, and eight declined to participate. To ensure the participant sample size was achieved, an additional 12 hotels were randomly selected with one declining to participate, resulting in a total of 30 participating hotels. All employees at selected hotels were eligible to participate. This permitted the inclusion of a wide array of hospitality worker types (e.g., maintenance workers, cleaning staff, waiters, reception clerks, etc.), despite the "cost" of higher intra-cluster similarity.

Prior to enrollment, we conducted in-person sensitization at each participating site. We enrolled participants over a 12 day period (June 13–24, 2022). Government workers were enrolled in the locations designated by the San Pedro Town Council, the seat of government in Ambergris Caye, or at the office where they work. We also offered enrollment to participants at a community-wide health fair and set up a dedicated enrollment site at a public location in the center of town on two separate days for those who were unavailable at the hotel or government workplace on their designated day of enrollment.

### Eligibility Criteria

Eligible individuals were (1) current government or tourism employees; (2) 18 years of age or older; (3), living in Ambergris Caye for at least three months; and (4) planned to continue living in the community for at least another six months. Participants were excluded if they declined to provide informed consent or were unwilling or unable to provide a blood sample.

### Data Collection

The WHO UNITY survey tool from the protocol "Population-based age-stratified seroepidemiological investigation protocol for coronavirus 2019 (COVID-19) infection" was used to collect demographic information and COVID-19 exposure,

symptom, and complications history [10]. We further expanded this protocol to include medical history, COVID-19 vaccination history, and epidemiological exposures. The survey tool was piloted with Belize-based staff and optimized for understandability based on feedback. The final version of the survey tool used in the study can be accessed through the Harvard Dataverse public repository at: https://doi.org/10.7910/DVN/BX5HUU [11]. Once participants were screened for eligibility and consented into the study, study staff interviewed each participant using the survey tool. The face-to-face interview optimized the interpretation of questions and maximized the completeness of data. Survey responses were collected on paper forms then entered into Research Electronic Data Capture (REDCap, Vanderbilt University, Nashville, TN USA). Participants were offered a small food treat for survey completion. All data were collected in English (official language of Belize) with interpretation support for Spanish-speaking personnel when needed.

## Laboratory procedures

During enrollment, blood was collected into serum separator tubes using standard venipuncture techniques. All samples were processed within two hours of collection, aliquoted, stored on ice packs, and transported to the Central Medical Laboratory in Belize City within 24 hours of collection. Serum samples were tested for the presence of SARS-CoV-2 specific IgG and total Ig antibodies using commercial ELISA kits authorized by the U.S. Food and Drug Administration under Emergency Use Authorization. We used the InBios SCoV-2 Detect IgG ELISA kit to detect IgG antibodies targeting the spike protein antigen, indicating prior natural infection and/or vaccination, with a reported sensitivity and specificity of 100% (InBios International, Inc., Seattle, Washington USA) [12]. We also used the Bio-Rad Platelia SARS-CoV-2 Total Ab Assay kit which detects IgM, IgG, and/or IgA antibodies targeting the nucleocapsid antigen (Bio-Rad Laboratory, Inc, Hercules, California USA), with a reported sensitivity and specificity of 98% and 99.3%, respectively [13]. Testing was performed according to manufacturers' protocols.

Positive detection of anti-nucleocapsid antibodies was used as a marker for prior infection among individuals that have either never received a COVID-19 vaccine, or among individuals that received spike-only based COVID-19 vaccines (Fig 1). Given that almost all COVID-19 vaccines administered in Belize prior to our study targeted immunogenicity against spike protein [14], we were able to use anti-nucleocapsid-based assays to assess prior natural infection [15]. Most antibody test results were delivered to the participants in-person, while others preferred via email or WhatsApp. Participants were invited to add their results to the national electronic medical record system (Belize Health Information System) if they were interested in discussing the results with their medical provider.

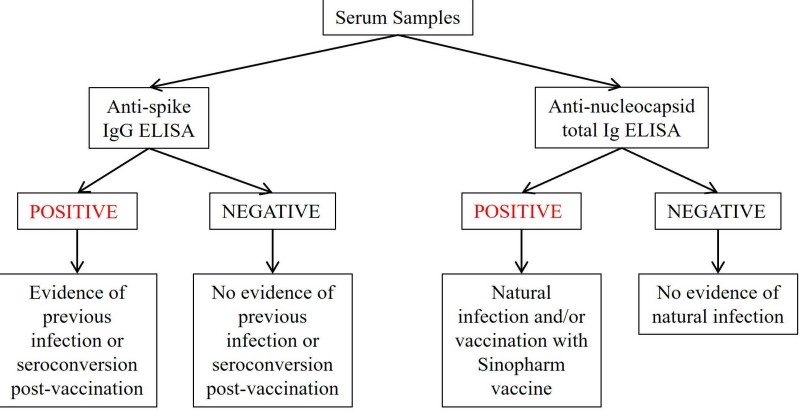

**Fig 1. Testing algorithm for community-based seroprevalence in Belize.**

## Statistical analyses

All statistical calculations were performed in Stata V18 (StataCorp LLC, College Station, TX USA) and NCSS 2025 (NCSS, Inc, Kaysville, UT USA). Graphs were constructed using GraphPad (GraphPad Software, UK). Descriptive statistics were used to calculate proportions. Simple logistic regression was used to identify risk factors, signs, and symptoms associated with SARS-CoV-2 infection or self-reported illness experienced since March 2020. Multivariable logistic regression models were constructed to identify independent factors associated with nucleocapsid positivity. Missing variables were rare (<1% across all variables); therefore, analyses were conducted using complete cases, and observations with missing values in any model variable were excluded. A backward stepwise approach was used for constructing the multivariable models; variables with $p < 0.25$ from the bivariate analyses were included. Variables with $p < 0.05$ (alpha = 0.05) were considered significant and were retained in the final model. The database has been made publicly available through the Harvard Dataverse public repository at: https://doi.org/10.7910/DVN/BX5HUU [11].

## Results

### Overall participant population

During the enrollment period, 572 potential participants were approached, and 561 agreed to participate, leading to a high participation rate of 98% (Fig 2). Of the 561 participants, 11 were excluded from analysis for the following reasons: three did not have a blood sample collected for testing, six reported not working in tourism or government sector, and one was found to be under 18 years (17.9 years) of age. Therefore, 551 participants were included in the analysis.

The median age of participants was 34.5 years (range 18–74 years), and 267 (48%) were female, including 5 who were pregnant. Most participants (n = 428, 78%) worked in the tourism sector, while 123 (22%) worked in government. A high proportion of the tourism sector participants worked in hotel maintenance, landscaping, cleaning or laundry (38%), followed by restaurant workers (25%), front desk or concierge workers (18%), management and administrative staff (16%), security (2%), tour operators (2%), public services and administration (0.2%), and medical (0.2%). Fourteen tourism workers (3%) listed more than one job function.

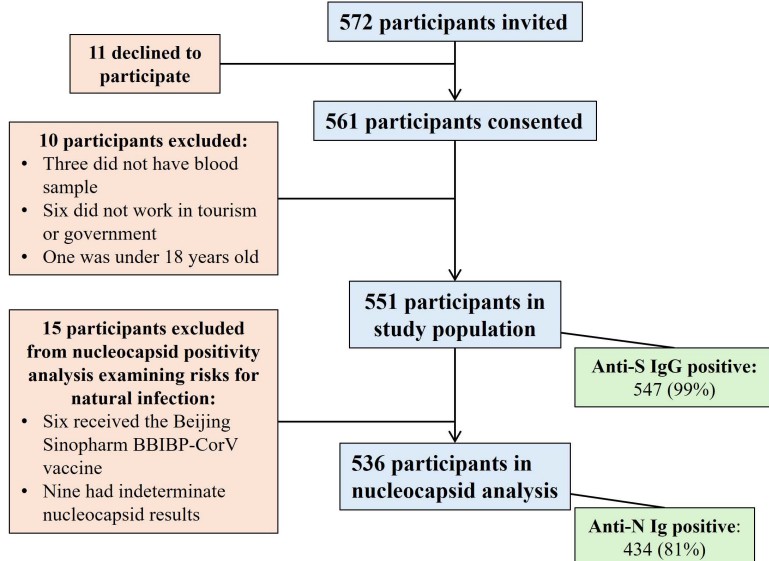

**Fig 2. Study population and sample for analysis.**

Of the 180 government workers, a high percentage (n = 123, 68%) participated in the study. Half (50%) worked in public services or administration, followed by schools (28%), law enforcement or security (15%), maintenance or cleaning (7%), emergency services (2%), or tour operation (2%). Five government workers (4%) reported more than one job function.

## SARS-CoV-2 spike protein seropositivity and vaccination history

Of the 551 participants, almost all (n = 547; 99%) were seropositive for IgG against the SARS-CoV-2 spike protein. Among those positive for the spike protein, almost all (539/547; 99%) reported a history of COVID-19 vaccination. Among the four participants who were spike protein negative, one reported no history of COVID-19 vaccination, and the remaining three (75%) reported a history of vaccination with AstraZeneca COVIDSHIELD, including one who had received a single dose and two who received two doses.

Most participants (98%) reported receiving one or more COVID-19 vaccine dose(s), and more than one-third (36%) had received the initial series plus a booster dose. For the first dose, most participants (84%) were vaccinated with AstraZeneca COVISHIELD (recombinant adenovirus encoding spike protein antigen), while 5% received Pfizer-BioNTech mRNA vaccine (spike protein encoding), 4% received Janssen/Johnson & Johnson (recombinant Ad26 vector encoding spike protein), 1% received Beijing Sinopharm BBIP-CorV (whole inactivated virus), 0.6% received Moderna (spike protein encoding), and the remaining 5% were unsure of vaccine brand. Only a few participants (n = 8, 1%) were unvaccinated, citing lack of time or money to get to an establishment offering vaccines (n = 1), concern over side effects (n = 2), concern over whether the vaccine would protect them (n = 1), pregnancy and breastfeeding (n = 2), religious or personal beliefs (n = 2), and fear of needles (n = 1).

Among participants who reported receiving at least one dose of COVID-19 vaccine, 62% (335/542) reported side effects. Among those who reported a side effect, the most common were fever (51%), soreness at injection site (47%), tiredness/fatigue/drowsiness (32%), muscle/body aches (30%), headache (27%) chills (24%), and joint/back/bone pain (14%). Most participants who had experienced side effects (94%) reported that symptoms had resolved by the time of enrollment, and most characterized their side effects as mild (63%) and lasting less than two days (67%).

The majority of participants who knew the date of their first dose (n = 427/501; 85%) were vaccinated between March and July 2021, which coincided with vaccination campaigns in Belize (Fig 3) [16]. Based on national data (https://covid19.who.int/data), there was a wave of reported COVID-19 cases in Belize before vaccines were available and again in late 2021 and early 2022; most participants were vaccinated prior to the larger second wave of cases.

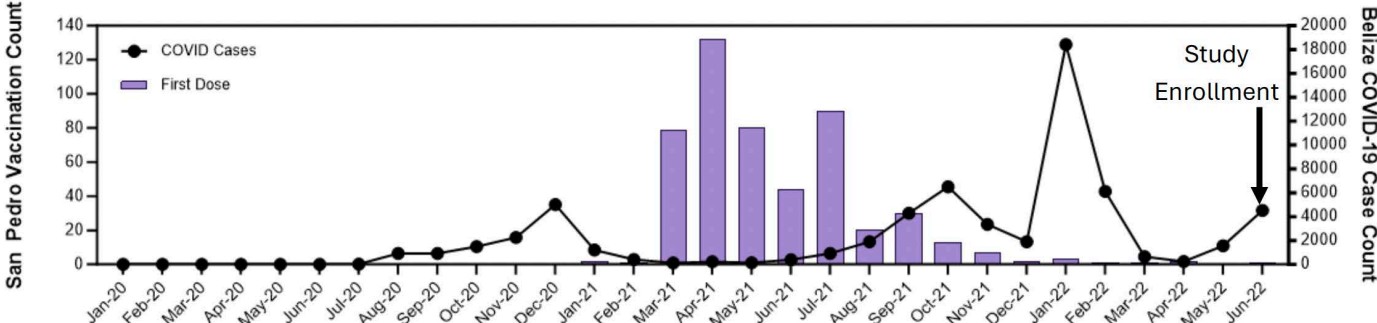

**Fig 3. Timing of first COVID-19 vaccination (month/year) of enrolled study participants compared to overall COVID-19 cases reported in Belize based on public data.** The black line represents nationally reported COVID-19 cases aggregated monthly and available from https://covid19.who.int/data. The purple bars represent the date of first COVID-19 vaccine dose among the study participants (n = 501, 91%).

## Prevalence of natural infection (nucleocapsid positivity) with SARS-CoV-2

A high proportion of participants (440/551, 80%) were nucleocapsid positive, 102 (19%) were nucleocapsid negative, and 9 (2%) had indeterminate results. All six participants who had received the Beijing Sinopharm BBIP-CorV vaccine were nucleocapsid positive, which is expected with administration of a whole inactivated virus vaccine which includes antigens to both spike and nucleocapsid proteins. These individuals were excluded from further statistical analysis. The 9 individuals with indeterminate results were also excluded, yielding a total sample of 536 for nucleocapsid analysis, with 434 (81%) seropositive for nucleocapsid antibodies and 102 (18%) seronegative.

The median age of nucleocapsid-negative participants was slightly but significantly older (36 vs 33 years) (Table 1). Females had significantly higher odds of being nucleocapsid positive compared to males (OR=1.56, p = 0.047). Roughly equal proportions of both nucleocapsid positive (n = 340, 78%) and nucleocapsid negative participants (n = 77, 75%) worked in the tourism sector, indicating employment type was not significantly associated with nucleocapsid positivity.

## Exposures to COVID-19

Most participants (78%) reported working in a job that most of the time required close contact (<6 feet) with guests or with the public in the six months prior to enrollment (Table 1), and most participants (70%) also reported direct contact (<6 feet) with a known COVID-19 positive individual during the time they would have been infectious; however, neither of these variables were significantly associated with being nucleocapsid positive.

Less than one-third of participants (29%) reported attending events with ≥10 people outside of work on a daily or weekly basis in the prior 6 months, and approximately one-third (36%) of participants reported attending or having someone in the family attend religious services in the prior six months. Almost a quarter of the participants (23%) reported living with ≥5 people in their home. Slightly over half (57%) reported having ≥2 essential workers in the home, and less than half (40%) reported having children who attended school or received childcare outside of the home in the prior year. About half (48%) of participants reported always or usually wearing a mask when outside of home, and slightly more than half (56%) reported always or usually carrying hand sanitizer. Living with ≥5 people in the household significantly increased the odds of nucleocapsid positivity (OR=2.15, p = 0.01).

## COVID-19 vaccination, testing, and illness history

Those who received a COVID-19 booster dose were significantly less likely to be nucleocapsid positive (OR=0.55, p = 0.01). About half of participants (51%) reported receiving a positive test for SARS-CoV-2 infection. Most of those who tested nucleocapsid positive (70%; 152/218) reported a positive test within 3–12 months prior to the date of study enrollment. A prior positive test for SARS-CoV-2 was significantly associated with nucleocapsid positivity (OR=2.70, p < 0.001). Interestingly, 27 of the 102 nucleocapsid seronegative participants (26%) reported a prior positive test for SARS-CoV-2, with three testing positive within a month of enrollment and 15 reporting a positive test >6 months prior to enrollment.

Half of the participants (n = 282; 53%) reported experiencing COVID-19-like symptoms since March 2020, with a higher percentage (55%) among nucleocapsid positive participants (OR=1.53, p = 0.054). Among all participants reporting any COVID-like symptoms, the most common were fever (65%), sore throat (53%), nasal congestion or runny nose (41%), and a new or unusual headache (40%) (Table 2). No individual symptom was significantly associated with nucleocapsid positivity.

Almost half of the participants who reported COVID-19/COVID-like illness (n = 129, 46%) sought medical care for their symptoms. Eight were hospitalized (8/129, 6%), and all were nucleocapsid positive. The median age of those hospitalized was 28, while the median age for participants who sought medical care for their symptoms was older (33 years). The eight participants were hospitalized for a median of five days (range 1–10 days). One required oxygen, and two were in intensive care. Only one hospitalized participant (age 31) reported a pre-existing medical condition (lung disease).

**Table 1. Risk factors for infection with SARS-CoV-2 using nucleocapsid positivity as a surrogate biomarker of infection.**

| Variable | All (N = 536) (%) | Nucleocap-sid negative (n = 102) | Nucleocap-sid positive (n = 434) | Odds Ratio (95% CI) outcome = nucleocapsid positive | P-value |
|---|---|---|---|---|---|
| **Demographics** | | | | | |
| Median age in years (range) | 34 (18 – 74) | 36 (18 – 74) | 33 (18 – 72) | 0.98 (0.96, 0.99) | **0.04** |
| Sex (Female) | 263 (49) | 41 (40) | 222 (51) | 1.56 (1.01, 2.42) | **0.047** |
| Tourism worker | 417 (78) | 77 (75) | 340 (78) | 1.17 (0.71, 1.95) | 0.53 |
| Government worker | 119 (22) | 25 (25) | 94 (22) | 0.85 (0.51, 1.41) | 0.53 |
| **Exposures to COVID-19** | | | | | |
| Job requires close contact (< 6 feet) with guests/public most of the time | 413/531 (78) | 85 (83) | 328/429 (76) | 0.65 (0.37, 1.14) | 0.14 |
| Had direct contact (< 6 feet) with known COVID-19 posi-tive person | 377/535 (70) | 72 (71) | 305/433 (70) | 0.99 (0.62, 1.59) | 0.98 |
| Attended large events regularly (daily or weekly) outside of work (with >10 people) in the prior 6 months | 154/535 (29) | 28 (27) | 126/433 (29) | 1.08 (0.67, 1.76) | 0.74 |
| Attended religious services outside of the home in the prior 6 months (participant or members of the household) | 194/533 (36) | 38 (37) | 156/431 (36) | 0.96 (0.61, 1.49) | 0.84 |
| Has five or more people in home | 124/534 (23) | 14 (14) | 110/432 (25) | 2.15 (1.17, 3.93) | **0.01** |
| Has two or more essential workers living in the home | 307 (57) | 53 (52) | 254 (59) | 1.30 (0.85, 2.01) | 0.23 |
| Had children in household who attended school or child-care outside the home in the prior year | 216/534 (40) | 36 (35) | 180/432 (42) | 1.31 (0.84, 2.05) | 0.24 |
| Always/ usually wore a mask when outside the home in the prior 6 months | 255 (48) | 47 (46) | 208 (48) | 1.08 (0.70, 1.66) | 0.74 |
| Always or usually carries hand sanitizer when leaving home | 298 (56) | 49 (48) | 249 (57) | 1.46 (0.94, 2.24) | 0.09 |
| **COVID-19 illness and vaccination** | | | | | |
| Ever tested positive for COVID-19 | 218/428 (51) | 27/85 (32) | 191/343 (56) | 2.70 (1.63, 4.47) | **<0.001** |
| Had COVID-19-related or COVID-19-like symptoms | 282/335 (53) | 45 (44) | 237/433 (55) | 1.53 (0.99, 2.36) | 0.054 |
| Received COVID-19 booster dose* | 196 (37) | 49 (48) | 147 (34) | 0.55 (0.36, 0.87) | **0.01** |
| Had side effects from the COVID-19 vaccine | 329/527 (62) | 61 (60) | 268/425 (63) | 1.15 (0.74, 1.79) | 0.54 |
| **Comorbidities** | | | | | |
| Had one or more pre-existing medical conditions | 86/533 (16) | 16 (16) | 70/431(16) | 1.04 (0.58, 1.88) | 0.89 |
| Diabetes | 27 (5) | 6 (6) | 21 (5) | 0.81 (0.32, 2.07) | 0.67 |
| Heart disease | 8 (1) | 1 (1) | 7 (2) | 1.66 (0.20, 13.61) | 0.64 |
| Asthma or other lung disease | 39 (7) | 5 (5) | 34 (8) | 1.65 (0.63, 4.33) | 0.31 |
| Immunocompromised | 11 (2) | 4 (4) | 7 (2) | 0.40 (0.12, 1.40) | 0.15 |

CI = confidence interval; *A COVID-19 booster is defined as the third dose if the participant received a two-dose initial vaccine series or the second dose if the participant received a one-dose Johnson & Johnson vaccine.

### Independent risk factors for SARS-CoV-2 infection

The following variables were entered into the backward stepwise multivariable logistic regression model to predict nucle-ocapsid positivity based on the results from the bivariate analysis: age (continuous), sex, job requires close contact with guests or the public, having ≥5 people in the home, having ≥2 essential workers in the home, children attending school/childcare outside the home, always or usually carrying hand sanitizer, history of a positive COVID-19 test, had COVID-19-related or COVID-19-like symptoms, received a COVID-19 booster dose, and a history of immune suppression. Since

**Table 2. Signs and symptoms reported by participants who had a history of COVID-19 or COVID-19-like illness.**

| Signs and symptoms | All (n = 282) (%) | Nucleocapsid negative (n = 45) | Nucleocapsid positive (n = 237) | Odds Ratio (95% CI) outcome = nucleocapsid positive | P-value |
|---|---|---|---|---|---|
| Fever | 182 (65) | 31 (69) | 151 (64) | 1.22 (0.77, 1.95) | 0.40 |
| New or worsening cough | 96 (34) | 19 (42) | 77 (32) | 0.94 (0.54, 1.64) | 0.83 |
| Shortness of breath/difficulty breathing | 61 (22) | 8 (18) | 53 (22) | 1.63 (0.75, 3.55) | 0.22 |
| Loss of taste or smell | 86 (30) | 12 (27) | 74 (31) | 1.54 (0.80, 2.96) | 0.19 |
| Sore throat | 150 (53) | 25 (56) | 125 (53) | 1.25 (0.76, 2.05) | 0.39 |
| Nasal congestion or runny nose | 117 (41) | 18 (40) | 99 (42) | 1.38 (0.79, 2.41) | 0.26 |
| New or unusual headache | 113 (40) | 21 (47) | 92 (39) | 1.04 (0.61, 1.77) | 0.89 |
| Chills | 76 (27) | 12 (27) | 64 (27) | 1.30 (0.67, 2.51) | 0.44 |
| Fatigue | 97 (34) | 17 (38) | 80 (34) | 1.13 (0.64, 2.01) | 0.68 |
| Nausea/vomiting | 29 (10) | 2 (4) | 27 (11) | 3.32 (0.78, 14.18) | 0.11 |
| Muscle aches | 100 (35) | 17 (38) | 83 (35) | 1.18 (0.67, 2.10) | 0.57 |
| Joint/body pain | 18 (6) | 3 (7) | 15 (6) | 1.18 (0.34, 4.16) | 0.80 |
| Chest pain | 5 (2) | 1 (2) | 4 (2) | 0.94 (0.10, 8.50) | 0.96 |
| Diarrhea | 26 (9) | 2 (4) | 24 (10) | 2.93 (0.68, 12.59) | 0.15 |
| Stomach pain lasting more than one day | 21 (7) | 1 (2) | 20 (8) | 4.88 (0.65, 36.79) | 0.12 |
| Rash related to an infection | 7 (2) | 0 (0) | 7 (3) | undefined | 0.92 |
| **Medical treatment/recovery** | | | | | |
| Sought medical care for symptoms | 129 (46) | 16 (36) | 113 (48) | 1.75 (0.90, 3.40) | 0.10 |
| Hospitalized for symptoms | 8 (3) | 0 (0) | 8 (3) | undefined | 0.97 |
| Recovered completely from illness | 240 (85) | 38 (84) | 202 (85) | 1.06 (0.38, 2.95) | 0.91 |

CI = confidence interval

receiving a history of a positive COVID-19 test is understandably associated with nucleocapsid positivity, we omitted this variable in a separate model and reran the analysis to understand independent risk factors for SARS-CoV-2 infection.

Based on multivariate modeling strategies, variables that were independently associated (p ≤ 0.05) with nucleocapsid positivity were having ≥5 people in the home (aOR=2.02), always or usually carrying hand sanitizer (aOR=1.69), history of a positive test for COVID-19 (aOR=2.72), and having received a COVID-19 booster dose (aOR=0.56) (Table 3). After removing the variable for testing positive for SARS-CoV-2, risk factors that were independently associated (p ≤ 0.05) with nucleocapsid positivity was having ≥5 people in the home (aOR=2.08), while receiving a COVID-19 booster dose remained protective (aOR=0.57).

## Discussion

We enrolled tourism and government workers from the island of Ambergris Caye, Belize to characterize SARS-CoV-2 seroprevalence in a region with a high influx of international visitors. We evaluated the seroprevalence of antibodies against the spike protein as a biomarker for vaccination, natural infection, or both, and antibodies against the nucleocapsid protein as a biomarker of prior natural infection [15]. Nearly all (99%) of our study participants had evidence of antibodies against the spike protein, which correlated to the high uptake of COVID-19 vaccination in this population (98%). Interestingly, a higher-than-expected percentage of the participants also had evidence of natural infection (81%), providing a high degree of hybrid immunity. Positivity related to natural infection was significantly associated with large households (≥5 members), prior positive COVID-19 test, and always/usually carrying hand sanitizer, while having a COVID-19 booster dose was found to be protective. When evaluating risk factors for natural SARS-CoV-2 infection, having ≥5 people in

PLOS Global Public Health

**Table 3. Multivariable logistic regression models with variables independently associated with nucleocapsid positivity and SARS-CoV-2 infection.**

| Variable | Model with variables independently associated with nucleocapsid positivity | | Model with variables independently associated with risk of SARS-CoV-2 infection | |
|---|---|---|---|---|
| | Adjusted OR (95% CI) | P-value | Adjusted OR (95% CI) | P-value |
| Has five or more people in home | 2.02 (1.05, 3.89) | 0.036 | 2.08 (1.13, 3.81) | 0.018 |
| Always or usually carries hand sanitizer when leaving home | 1.69 (1.03, 2.77) | 0.039 | – | – |
| Ever tested positive for COVID-19 | 2.72 (1.63, 4.55) | <0.001 | – | – |
| Received COVID-19 booster dose* | 0.56 (0.34, 0.93) | 0.024 | 0.57 (0.37, 0.89) | 0.013 |

OR = odds ratio; CI = confidence interval*A COVID-19 booster is defined as the third dose if the participant received a two-dose initial vaccine series or the second dose if the participant received a one-dose Johnson & Johnson vaccine.

home remained associated with infection, while having received a COVID-19 booster dose remained protective. Approximately half of all participants reported symptoms of COVID-19/COVID-like illness, and interestingly, 45% of those who were nucleocapsid positive reported no symptoms related to COVID-19 or a COVID-19-like illness. Even with the high percentage of positivity, hospitalization rates were very low, likely due to protections related to the high level of vaccination coverage in the population as almost all participants had received a primary dose prior to surge in COVID-19 cases in 2021 and 2022 after travel restrictions had been lifted.

We found that participants with ≥5 people in the household were at higher risk of infection. This is consistent with findings from a study based in the microstate of Andorra, a European tourist destination, where there was a higher seroprevalence (18%) after the first wave of the pandemic among those who reported living in a household with ≥6 members when compared to the overall seroprevalence in the population (11%) [17]. Another study in the Dominican Republic in 2021 found that risk of being seropositive for SARS-CoV-2 antibodies increased by 1.4- and 1.7-fold in households with 5–6 and ≥7 residents, respectively, compared to households with 1–2 residents [18]. A population-based seroprevalence study in Nicaragua demonstrated increased seropositivity in participants who reported a seropositive individual in their household [19]. These studies emphasize the importance of household transmission of COVID-19, with living in larger households increasing the risk for infection. Specific messaging on prevention and household risk reduction may be useful, particularly in areas where tourism sector workers often live in larger, shared households.

A surprising result was that participants who always/usually carried hand sanitizer were more likely to be seropositive for nucleocapsid antibodies. The WHO and Centers for Disease Control and Prevention (CDC) both recommend frequent handwashing with soap and water or using alcohol-based hand sanitizer as effective mechanisms for preventing the spread of COVID-19 [20, 21]. We hypothesize that carrying hand sanitizer may be an indication of the individuals' higher risk perception due to their work or other environment, which did in fact relate to the likelihood of infection. An alternative hypothesis is that carrying hand sanitizer can provide a false sense of security of protection against a respiratory virus like COVID-19. Additionally, carrying hand sanitizer might not necessarily correlate with usage.

As expected, individuals who reported a history of a positive COVID-19 test were more likely to be nucleocapsid positive. These findings support the validity of using assays that detect antibodies against the nucleocapsid antigen as a biomarker for natural infection, especially in populations primarily vaccinated with products that induce immunogenicity against the SARS-CoV-2 spike protein antigen. One limitation of this approach is that total Ig antibodies to the nucleocapsid protein wane over time to undetectable levels in naturally infected individuals, with sero-reversion estimates of 11% and 19% at one and two years post-infection, respectively [22, 23]. In our study, 20 participants who tested negative for nucleocapsid antibodies reported both a history of a positive COVID-19 test and a COVID-19-like illness since March 2020, hence we conjecture that the negative findings were most likely due to sero-reversion. These findings highlight the

importance of capturing data on the specific vaccine product administered when assessing vaccination history along with timing of COVID-19-like symptoms. It is interesting to note that symptoms reported by nucleocapsid-positive participants in our study were no different from symptoms reported by nucleocapsid-negative participants, highlighting the difficulties in distinguishing COVID-19 from other respiratory illnesses.

Protection for essential workers, such as those in tourism and government services, will be critical as we manage successive waves of SARS-CoV-2 and other emerging respiratory pathogens. Our study illustrates that most study participants, though vaccinated, also experienced natural infection, though with low reported incidence of severe disease. While strict movement restrictions were initially implemented, these were unsustainable for a country economically dependent on tourism. As of March 2023, 63% of the Belizean population had received at least one dose of the COVID-19 vaccine, which is above the global average [24]. Prior research has found hybrid immunity influenced by both vaccination and natural infection synergistically heightens the level and duration of protection [8, 9]. In addition to vaccination efforts, targeted and practical prevention approaches—such as tactics for reducing household transmission—will be important to increase protection for essential public workers.

There are several limitations worth noting. First, it was challenging to have a complete sampling frame, making the participant selection subject to bias. While helpful to assess clinical and vaccination history and potential risk factors for infection, there is a risk for recall and response bias. Participants might have felt influenced to respond positively to questions related to behaviors to prevent infections, including wearing masks and carrying hand sanitizer. This could contribute to our finding that carrying hand sanitizer was associated with nucleocapsid positivity. Another limitation worth noting is that we had a small number (n = 6) of participants who reported vaccination with Beijing Sinopharm's BBIP-CorV inactivated whole virus vaccine, which would then influence nucleocapsid antibody response from vaccination; therefore, these individuals were excluded from analysis. There were 29 participants who were unsure which vaccine they had received. While only 10,000 doses of Sinopharm vaccine were donated to Belize over the course of the pandemic [25], and only 1% of our participants in our study reported receiving this vaccine, it is possible some of these 29 participants received Sinopharm and should have been excluded from the nucleocapsid analysis. We felt this situation would have been unlikely, but nevertheless it could have influenced the statistical analysis.

## Conclusion

This study provides a unique opportunity to assess vaccination coverage and risk factors for COVID-19 infection among a high-risk population living and working in a geographic area with a high influx of travelers from all over the world. Vaccination uptake was high as evidenced by self-report and 99% prevalence of anti-SARS-CoV-2 spike protein antibodies. The high seroprevalence for nucleocapsid protein indicates that most of the study population has hybrid immunity resulting from prior natural infection and vaccination. This level of population immunity would likely reduce the risk of severe disease and workforce disruption given the likelihood of frequent occupational exposures. Tourism-sector workers represent a critical interface between global travel networks and the local communities in which they live. To our knowledge, this is the first COVID-19 seroprevalence study focused on the tourism industry in Central America. The findings from this study underscore the value of targeted vaccination and surveillance strategies in tourism-dependent communities and provide actionable evidence to inform public health preparedness and risk mitigation strategies for future pandemics.

## Acknowledgments

We would like to thank Drs. Emily Zielinkski-Gutierrez, Rafael Chacon, and Beatriz Lopez from the Centers for Disease Control and Prevention Central America Office for their guidance and support, and JongIn Hwang at Emory University for assisting with manuscript formatting. We would also like to thank our field team members for their hard work and dedication: Betty Hernandez, Ruby Aguilon, Amelita Jacobs, Kirk Lainfiesta, Amir Arana, Diosa Young, Dylan Yacab, Rosalva Blanco, and Roberto Melendez.

## Author contributions

**Conceptualization:** Russell Manzanero, Gerhaldine Morazan, Kristy O. Murray.

**Data curation:** Oluwadara Okeremi, Ella Hawes, Kristy O. Murray.

**Formal analysis:** Oluwadara Okeremi, Ella Hawes, Sarah M. Gunter, Shannon E. Ronca, Eric J. Nilles, Kristy O. Murray.

**Funding acquisition:** Kristy O. Murray.

**Investigation:** Oluwadara Okeremi, Ella Hawes, Anh N. Ly, Allison Lino, Allison Stewart-Ruano, Gerhaldine Morazan, Kristy O. Murray.

**Methodology:** Sarah M. Gunter, Shannon E. Ronca, Allison Stewart-Ruano, Gerhaldine Morazan, Kristy O. Murray.

**Project administration:** Oluwadara Okeremi, Anh N. Ly, Allison Lino, Allyson Hidalgo, Meghan Si, Allison Stewart-Ruano, Adriana Maliga, Gerhaldine Morazan, Kristy O. Murray.

**Resources:** Oluwadara Okeremi, Ella Hawes, Anh N. Ly, Russell Manzanero, Sarah M. Gunter, Shannon E. Ronca, Allison Lino, Allyson Hidalgo, Meghan Si, Allison Stewart-Ruano, Adriana Maliga, Eric J. Nilles, Gerhaldine Morazan, Kristy O. Murray.

**Supervision:** Gerhaldine Morazan, Kristy O. Murray.

**Validation:** Oluwadara Okeremi, Ella Hawes, Anh N. Ly, Sarah M. Gunter, Shannon E. Ronca, Gerhaldine Morazan, Kristy O. Murray.

**Visualization:** Oluwadara Okeremi, Ella Hawes, Sarah M. Gunter, Shannon E. Ronca, Kristy O. Murray.

**Writing – original draft:** Oluwadara Okeremi, Ella Hawes, Kristy O. Murray.

**Writing – review & editing:** Oluwadara Okeremi, Ella Hawes, Anh N. Ly, Russell Manzanero, Sarah M. Gunter, Shannon E. Ronca, Allison Lino, Allyson Hidalgo, Meghan Si, Allison Stewart-Ruano, Adriana Maliga, Eric J. Nilles, Gerhaldine Morazan, Kristy O. Murray.

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
