## [Decision Letter · Decision Letter 0]

4 Dec 2025

PGPH-D-25-02807

Hybrid SARS-CoV-2 Immunity Among Frontline Workers in a High-Tourism Setting: A Community-Based Serosurvey in Ambergris Caye, Belize, June 2022

Dear Dr. Murray,

Thank you for submitting your manuscript to PLOS Global Public Health. After careful consideration, we feel that it has merit but does not fully meet PLOS Global Public Health’s publication criteria as it currently stands. Therefore, we invite you to submit a revised version of the manuscript that addresses the points raised during the review process.

We have received some suggestions from our reviewers concerning the significance of the study and the availability of data that should be addressed mainly. Please revise it at your earliest convenience so we can make our next decision.

We look forward to receiving your revised manuscript.

Kind regards,

Kanokwan Suwannarong, Ph.D.

Academic Editor

Journal Requirements:

Please provide additional details regarding participant consent. In the ethics statement in the Methods and online submission information, please ensure that you have specified (1) whether consent was informed and (2) what type you obtained (for instance, written or verbal, and if verbal, how it was documented and witnessed). If your study included minors, state whether you obtained consent from parents or guardians. If the need for consent was waived by the ethics committee, please include this information.

Please include a complete copy of PLOS’ questionnaire on inclusivity in global research in your revised manuscript. Our policy for research in this area aims to improve transparency in the reporting of research performed outside of researchers’ own country or community. The policy applies to researchers who have travelled to a different country to conduct research, research with Indigenous populations or their lands, and research on cultural artefacts. The questionnaire can also be requested at the journal’s discretion for any other submissions, even if these conditions are not met. Please find more information on the policy and a link to download a blank copy of the questionnaire here: https://journals.plos.org/globalpublichealth/s/best-practices-in-research-reporting. Please upload a completed version of your questionnaire as Supporting Information when you resubmit your manuscript.

Please send a completed 'Competing Interests' statement, including any COIs declared by your co-authors. If you have no competing interests to declare, please state "The authors have declared that no competing interests exist".

Please amend your detailed Financial Disclosure statement. This is published with the article. It must therefore be completed in full sentences and contain the exact wording you wish to be published.a. State the initials, alongside each funding source, of each author to receive each grant. For example: "This work was supported by the National Institutes of Health (####### to AM; ###### to CJ) and the National Science Foundation (###### to AM)."

In the online submission form, you indicated that “Deidentified data are available upon request, contingent upon approval of a data sharing agreement.”All PLOS journals now require all data underlying the findings described in their manuscript to be freely available to other researchers, either1. In a public repository,2. Within the manuscript itself, or3. Uploaded as supplementary information.This policy applies to all data except where public deposition would breach compliance with the protocol approved by your research ethics board. If your data cannot be made publicly available for ethical or legal reasons (e.g., public availability would compromise patient privacy), please explain your reasons by return email and your exemption request will be escalated to the editor for approval. Your exemption request will be handled independently and will not hold up the peer review process, but will need to be resolved should your manuscript be accepted for publication. One of the Editorial team will then be in touch if there are any issues.Please provide separate figure files in .tif or .eps format.For more information about figure files please see our guidelines:
https://journals.plos.org/globalpublichealth/s/figures

Additional Editor Comments (if provided):

Dear Authors,

We appreciate your submission of this manuscript for our review. We have received some suggestions from our reviewers concerning the significance of the study and the availability of data that should be addressed mainly. Please revise it at your earliest convenience so we can make our next decision.

Kind regards,

Reviewers' comments:

Reviewer's Responses to Questions

**Comments to the Author**

1. Does this manuscript meet PLOS Global Public Health’s publication criteria? Is the manuscript technically sound, and do the data support the conclusions? The manuscript must describe methodologically and ethically rigorous research with conclusions that are appropriately drawn based on the data presented.? Is the manuscript technically sound, and do the data support the conclusions? The manuscript must describe methodologically and ethically rigorous research with conclusions that are appropriately drawn based on the data presented.

Reviewer #1: Yes

Reviewer #2: Yes

Reviewer #3: Yes

2. Has the statistical analysis been performed appropriately and rigorously?

Reviewer #1: Yes

Reviewer #2: Yes

Reviewer #3: Yes

3. Have the authors made all data underlying the findings in their manuscript fully available (please refer to the Data Availability Statement at the start of the manuscript PDF file)?

The PLOS Data policy requires authors to make all data underlying the findings described in their manuscript fully available without restriction, with rare exception. The data should be provided as part of the manuscript or its supporting information, or deposited to a public repository. For example, in addition to summary statistics, the data points behind means, medians and variance measures should be available. If there are restrictions on publicly sharing data—e.g. participant privacy or use of data from a third party—those must be specified.requires authors to make all data underlying the findings described in their manuscript fully available without restriction, with rare exception. The data should be provided as part of the manuscript or its supporting information, or deposited to a public repository. For example, in addition to summary statistics, the data points behind means, medians and variance measures should be available. If there are restrictions on publicly sharing data—e.g. participant privacy or use of data from a third party—those must be specified.

Reviewer #1: No

Reviewer #2: Yes

Reviewer #3: No

4. Is the manuscript presented in an intelligible fashion and written in standard English?

Reviewer #1: Yes

Reviewer #2: Yes

Reviewer #3: Yes

5. Review Comments to the Author

Reviewer #1: **Recommendation: Extensive Revisions**

While the manuscript is methodologically sound, ethically rigorous, and well-written, the critical issue of **data availability** necessitates **extensive revisions**. The current statement regarding data availability ("available upon request, contingent upon approval of a data sharing agreement") does not comply with PLOS Global Public Health’s policy for unrestricted data sharing. The authors must either:

1. Deposit all de-identified raw data in a publicly accessible, recognized repository and provide the corresponding access link (DOI) in the manuscript.

2. Provide a compelling justification for any restrictions that fully aligns with the "rare exception" clause of the PLOS data policy, along with a transparent mechanism for data access that minimizes barriers.

In addition to this primary concern, I recommend addressing the following minor points to further enhance the manuscript's quality during revision:

* **Questionnaire Psychometric Properties:** In the "Data Collection" section, although the "standardized survey" is mentioned, there is no discussion of its psychometric properties (e.g., how content validity was ensured, if it was piloted, or if it's an adaptation of a validated tool). A brief clarification on the development or adaptation process of the questionnaire would strengthen the methodological description.

* **Clarity on "Standardized Survey" Source:** While it mentions being based on the WHO UNITY protocol, explicitly state whether the survey was directly adopted, adapted, or newly developed for this study, providing a reference for the original if applicable.

These revisions, particularly regarding data availability, are crucial for the manuscript to meet all publication standards for PLOS Global Public Health.

Reviewer #2: The manuscript describes a well-executed study with strong laboratory methods and addressing a relevant and interesting public health topic. However, certain required elements are unclear and revision is recommended to improve the clarity and transparency of the study.

The Introduction does not adequately explain why studying hybrid immunity is important in the current context. Since this is central theme of the paper, a clearer rationale would strengthen the background. Similarly, the choice of tourism workers as the study population needs a but more justification. Although the objectives are stated, they do not consistently align with the results.

The study is clearly a cross-sectional study, but it is not stated in the methodology or even the title or the abstract. The Sampling approach seems a bit complex. The description of the cluster sampling can be explained more clearly considering some selection bias. The exact recruitment period and duration of the data collection is also not mentioned anywhere. Sample Size is unclear, especially with different denominators across variables. While statistical analysis is strong, the manuscript should mention how missing data were handled.

The descriptive data given in tables and also neatly explained in the results section, but each table has a lot of variables, makes it slightly difficult to interpret. The prevalence of the nucleocapsid positivity which is the natural infection is reported also the anti spike IgG ELISA positivity is reported but the hybrid immunity is not interpreted.

The Discussion section contains valid points but not well organized and doesn't relate with study title and objectives and lacks clear narrative on hybrid immunity which is the theme of the study.

The conclusion does not fully answer all objectives and fails to emphasize on the implications for public health and tourism-sector risk.

A uniform style of referencing to be followed throughout, the text should use superscript numerical citations as per Vancouver style of referencing.

Overall, This is a good study with strong tehnical work and it would benefit from clearer reporting. The study needs to align with STROBE guidelines to improve clarity, and build coherence between objectives, methods, results and conclusions.

Reviewer #3: The research is relevant to PLOS Global Public Health and it was well written. However, I could not see the statement on the data availability/accessibility for the study. In addition, the abstract should have been structured as it is better for presentation and readership. Conflict of interest was not declared in the study.

6. PLOS authors have the option to publish the peer review history of their article (what does this mean?). If published, this will include your full peer review and any attached files.). If published, this will include your full peer review and any attached files.

**Do you want your identity to be public for this peer review?** For information about this choice, including consent withdrawal, please see our Privacy Policy..

Reviewer #1: **Yes:** JÚLIO CÉSAR ANDRÈJÚLIO CÉSAR ANDRÈ

Reviewer #2: **Yes:** Upasana Raj BattinaUpasana Raj Battina

Reviewer #3: **Yes:** Olumuyiwa Elijah AriyoOlumuyiwa Elijah Ariyo

 Figure Resubmissions:

---

## [Decision Letter · Decision Letter 1]

15 Mar 2026

Hybrid SARS-CoV-2 Immunity Among Frontline Workers in a High-Tourism Setting: A Community-Based Serosurvey in Ambergris Caye, Belize, June 2022

PGPH-D-25-02807R1

Dear Dr. Murray,

We are pleased to inform you that your manuscript 'Hybrid SARS-CoV-2 Immunity Among Frontline Workers in a High-Tourism Setting: A Community-Based Serosurvey in Ambergris Caye, Belize, June 2022' has been provisionally accepted for publication in PLOS Global Public Health.

Best regards,

Kanokwan Suwannarong, Ph.D.

Academic Editor

Reviewer Comments (if any, and for reference):

Reviewer's Responses to Questions

**Comments to the Author**

1. If the authors have adequately addressed your comments raised in a previous round of review and you feel that this manuscript is now acceptable for publication, you may indicate that here to bypass the “Comments to the Author” section, enter your conflict of interest statement in the “Confidential to Editor” section, and submit your "Accept" recommendation.

Reviewer #2: All comments have been addressed

Reviewer #3: All comments have been addressed

2. Does this manuscript meet PLOS Global Public Health’s publication criteria? Is the manuscript technically sound, and do the data support the conclusions? The manuscript must describe methodologically and ethically rigorous research with conclusions that are appropriately drawn based on the data presented.? Is the manuscript technically sound, and do the data support the conclusions? The manuscript must describe methodologically and ethically rigorous research with conclusions that are appropriately drawn based on the data presented.

Reviewer #2: Yes

Reviewer #3: Yes

3. Has the statistical analysis been performed appropriately and rigorously?

Reviewer #2: Yes

Reviewer #3: Yes

4. Have the authors made all data underlying the findings in their manuscript fully available (please refer to the Data Availability Statement at the start of the manuscript PDF file)?

The PLOS Data policy requires authors to make all data underlying the findings described in their manuscript fully available without restriction, with rare exception. The data should be provided as part of the manuscript or its supporting information, or deposited to a public repository. For example, in addition to summary statistics, the data points behind means, medians and variance measures should be available. If there are restrictions on publicly sharing data—e.g. participant privacy or use of data from a third party—those must be specified.requires authors to make all data underlying the findings described in their manuscript fully available without restriction, with rare exception. The data should be provided as part of the manuscript or its supporting information, or deposited to a public repository. For example, in addition to summary statistics, the data points behind means, medians and variance measures should be available. If there are restrictions on publicly sharing data—e.g. participant privacy or use of data from a third party—those must be specified.

Reviewer #2: Yes

Reviewer #3: Yes

5. Is the manuscript presented in an intelligible fashion and written in standard English?

Reviewer #2: Yes

Reviewer #3: Yes

6. Review Comments to the Author

Reviewer #2: All the queries have been addressed. This paper looks ready for submission.

Reviewer #3: The research is relevant and the manuscript is well written. Previous comments have been addressed satisfactorily

7. PLOS authors have the option to publish the peer review history of their article (what does this mean?). If published, this will include your full peer review and any attached files.). If published, this will include your full peer review and any attached files.

**Do you want your identity to be public for this peer review?** For information about this choice, including consent withdrawal, please see our Privacy Policy..

Reviewer #2: **Yes:** Battina Upasana RajBattina Upasana Raj

Reviewer #3: **Yes:** Olumuyiwa Elijah ARIYOOlumuyiwa Elijah ARIYO
